# On Exact Bit-level Reversible Transformers Without Changing Architecture

Guoqiang Zhang [1]   J.P. Lewis [2]   W. Bastiaan Kleijn [3]

## Abstract

In this work we present the BDIA-transformer, which is an exact bit-level reversible transformer that uses an unchanged standard architecture for inference. The basic idea is to first treat each transformer block as the Euler integration approximation for solving an ordinary differential equation (ODE) and then incorporate the technique of bidirectional integration approximation (BDIA) (originally designed for diffusion inversion) into the neural architecture, together with activation quantization to make it exactly bit-level reversible. In the training process, we let a hyper-parameter $\gamma$ in BDIA-transformer randomly take one of the two values $\{0.5, -0.5\}$ per training sample per transformer block for averaging every two consecutive integration approximations. As a result, BDIA-transformer can be viewed as training an ensemble of ODE solvers parameterized by a set of binary random variables, which regularizes the model and results in improved validation accuracy. Lightweight side information is required to be stored in the forward process to account for binary quantization loss to enable exact bit-level reversibility. In the inference procedure, the expectation $\mathbb{E}(\gamma) = 0$ is taken to make the resulting architecture identical to transformer up to activation quantization. Our experiments in natural language generation, image classification, and language translation show that BDIA-transformers outperform their conventional counterparts significantly in terms of validation performance while also requiring considerably less training memory. Thanks to the regularizing effect of the ensemble, the BDIA-transformer is particularly suitable for fine-tuning with limited data. Source-code can be found via this link.

[1]Department of Computer Science, University of Exeter, UK [2]NVDIA, USA [3] School of Engineering and Computer Science, Victoria University of Wellington, New Zealand. Correspondence to: Guoqiang Zhang <guoqiang.x.zhang@gmail.com>.

*Proceedings of the 42nd International Conference on Machine Learning*, Vancouver, Canada. PMLR 267, 2025. Copyright 2025 by the author(s).

## 1. Introduction

An active research trend in deep learning is to scale up the size of both the deep neural networks (DNNs) and the training data, with the aim of obtaining universal machine-learning models that are capable of accomplishing various tasks. Examples are large language models (LLMs) such as GPT-4 (Achiam et al., 2023) and Llama 2 (Touvron et al., 2023), which can, for example, have informative and friendly conversations with humans, solve mathematical problems, and produce high-quality source codes for programming tasks. A major bottleneck for training those large DNNs is that they often require large on-chip memory and large inter-chip communication bandwidth across many chips to accommodate both the DNN model and the intermediate activations of the input data-batch in back-propagation (Gholami et al., 2024), which is referred to as *memory wall* in the literature.

One promising technique to alleviate the issue of memory wall is to design and train reversible DNNs (Dinh et al., 2014; 2016; Behrmann et al., 2019; Mangalam et al., 2023). By doing so, the intermediate activations in the forward pass do not have to be stored in the memory to allow for back-propagation. Instead, they can be recomputed on-the-fly in the backward pass by exploiting the reversibility of the DNN, thus saving memory consumption by a large margin for deep DNNs. The procedure essentially reduces memory consumption at the cost of a reasonable amount of additional computation. The reduction in memory also has the potential to improve throughput by increasing the batch size.

Early reversible DNNs, such as NICE (Dinh et al., 2014), the methods of (Rezende & Mohamed, 2015), Real NVP (Dinh et al., 2016), and Inverse Autoregressive Flow (Kingma et al., 2016) are differentiable transformations that were aimed at facilitating generative modeling. For an overview of such early normalizing flows see (Kobyzev et al., 2020). Inspired by these works, a range of reversible residual models were proposed subsequently, including RevNet (Gomez et al., 2017), Glow (Kingma & Dhariwal, 2018), i-RevNet (Jacobsen et al., 2018), i-ResNet (Behrmann et al., 2019), layerwise inversion (Hascoet et al., 2019), Fourier-transformation based CNN inversion (Finzi et al., 2019), and Mintnet (Song et al., 2019). Another line of research enforces reversibility in deep learning from the perspec-

tive of ordinary differential equations (ODEs), which includes FFJORD (Grathwohl et al., 2018), leapfrog networks (Chang et al., 2017), momentum residual networks (Sander et al., 2021), and neural ODE inversion (Stam, 2022).

Recently, the research on reversibility has moved to other types of neural networks. The authors in (Mangalam et al., 2023; Zhu & Mangalam, 2023) proposed reversible vision transformers (referred to as RevViT) due to the popularity of LLMs. (Wallace et al., 2023) utilized a reversible diffusion sampling method for the task of diffusion based image editing. To our best knowledge, all the above existing reversible DNNs either require non-standard architectures or are constructed by modifying the original DNN architectures considerably to enable reversibility.

In this paper, we propose BDIA-transformer, a new type of reversible transformer that uses an unchanged, standard architecture for the inference procedure. It is based on the bidirectional integration approximation (BDIA), which was recently proposed in (Zhang et al., 2023a) to enable diffusion inversion for round-trip image editing. To be able to incorporate BDIA into transformers for online back-propagation, we follow the common practice of treating each transformer block as an Euler integration approximation for solving an ordinary differential equation (ODE).

We make two main contributions in this work. Firstly, we propose BDIA-transformers by introducing a random hyperparameter $\gamma \in \{-0.5, 0.5\}$ per transformer block per training sample to regularize the DNN models for improvement of validation performance. Each $\gamma$ parameter intends to average every two consecutive integration approximations. The training procedure becomes training of an ensemble of ODE solvers parameterized by a set of binary random variables. In the inference procedure, the expectation $\mathbb{E}[\gamma] = 0$ is utilized, which reduces BDIA-transformers to conventional transformers. As a result, our method is a good candidate for fine-tuning existing transformer-based models such as LLMs.[1]

Secondly, we perform activation quantization to allow for exact bit-level reversibility of BDIA-transformers. Note that the special setup of the $\gamma$ values in the subset $\{-0.5, 0.5\}$ when performing activation quantization leads to a 1 bit information loss per activation value per transformer block. Therefore, lightweight side information per transformer block needs to be stored during training to recover this 1 bit information loss. Despite this, the overall memory use is significantly reduced.

Experimental results for natural language generation (NLG), image classification, and language translation show that the BDIA technique significantly improves the validation perfor-

---

[1]See Subsection 5.1 for full fine-tuning and Appendix E for LoRA-based fine-tuning.

mance over that of the corresponding baseline transformers and simultaneously reduces training memory significantly. The improved performance results from the model regularization imposed by a set of $\gamma$ random variables. Our empirical study also indicates that RevViT from (Mangalam et al., 2023) produces either inferior or comparable validation performance to that of its original counterparts.

## 2. Related Works

In recent years, various quantization strategies (Yang et al., 2019; Wu et al., 2018; Wang et al., 2018) have been proposed in the training and/or inference processes of DNN models on low-precision devices. For instance, the work (Wu et al., 2018) successfully performed quantization on DNN weights, activations, gradients and errors in the training and inference processes and obtained promising results. The recent work (Ma, 2024) demonstrated that LLMs with a quantization of 1.58 bits per model parameter exhibit comparable performance to non-quantized models. In summary, it was found that the validation performance of DNN models that incorporate those quantization operations is comparable with that of conventional DNN models. In our work, we only need to apply activation quantization to enable exact bit-level reversibility in training BDIA-transformers.

We note that in general, model quantization and design of reversible DNN models are two complimentary strategies for reducing memory consumption in the training process, where the first strategy operates on the model parameters and the second one operates on the activation values. With the development of new model quantization methods such as (Ma, 2024), advancing the research frontier of reversible DNN models has become of great interest.

In addition to reversible DNNs, other memory-saving techniques have been proposed in recent years, such as gradient checkpointing (Feng & Huang, 2021), and activation offloading (Wu et al., 2024). At least in principle, the above techniques can be combined with the BDIA algorithm to save training memory for a broad class of residual-type neural architectures.

## 3. Preliminary

**Neural networks as ODEs**: (Chen et al., 2018) highlighted the interpretation that passing a hidden state across layers that add a correction to that state can be viewed as Euler integration. While that paper identified architectures including residual nets and normalizing flows as following this pattern, it is equally true of diffusion models and transformers. This interpretation emphasizes the importance of considering more accurate integration schemes (Karras et al., 2022). The need for improved integration is particularly apparent in round-trip image editing in diffusion models, where significant integration error will result in unintended visible

alterations to the image.

**Diffusion sampling via solving ODE**: Recently, the work (Zhang et al., 2023a) proposed the BDIA technique to enable diffusion inversion for effective round-trip image editing. From a high level point of view, BDIA can be viewed as a time-reversible ODE solver. Given an initial diffusion state $z_T$ at time step $T$, the diffusion-based sampling process for generating realisic images $z_\epsilon$ at time $t = \epsilon > 0$ can be realized by solving a probability ordinary differential equation (ODE)

$$dz = d(z, t)dt \tag{1}$$

over the time interval $t \in [T, \epsilon]$. The gradient vector $d(z, t)$ includes the output of a pre-trained DNN model with $(z_t, t)$ as its input. The common practice for solving the above ODE is to first discretize the continuous time interval $[T, \epsilon]$ properly into a set of timesteps $\{t_i | i = 0, \ldots, N\}$ with $t_0 = T$ and $t_N = \epsilon$, and then perform certain integration approximation per small time-interval sequentially to compute the final diffusion state $z_N = z_\epsilon$.

**BDIA**: Suppose we would like to estimate the next diffusion state $z_{i+1} = z_{t_{i+1}}$ by solving (1) based on the recent information $(z_i, t_i)$ and $(z_{i-1}, t_{i-1})$, where $z_j = z_{t_j}$ for $j = i - 1, i$. The basic idea of BDIA is to compute $z_{i+1}$ by performing both the forward integration approximation $\Delta(t_i \rightarrow t_{i+1}|z_i) \left(\approx \int_{t_i}^{t_{i+1}} d(z, t)dt\right)$ and the backward integration approximation $\Delta(t_i \rightarrow t_{i-1}|z_i)$ $\left(\approx -\int_{t_{i-1}}^{t_i} d(z, t)dt\right)$ conditioned on $z_i$. One popular method for implementing $\Delta(t_i \rightarrow t_{i+1}|z_i)$ and $\Delta(t_i \rightarrow t_{i-1}|z_i)$ in the literature of diffusion models is by employing the DDIM update expression (see (Song et al., 2021; Zhang et al., 2023b;a) for details). With the above two integration approximations, $z_{i+1}$ can be expressed as

$$z_{i+1} = z_{i-1} \underbrace{-(1 - \gamma)(z_{i-1} - z_i) - \gamma\Delta(t_i \rightarrow t_{i-1}|z_i)}_{\approx \int_{t_{i-1}}^{t_i} d(z,t)dt}$$

$$+ \underbrace{\Delta(t_i \rightarrow t_{i+1}|z_i)}_{\approx \int_{t_i}^{t_{i+1}} d(z,t)dt} \tag{2}$$

$$= \gamma z_{i-1} + (1 - \gamma)z_i - \gamma\Delta(t_i \rightarrow t_{i-1}|z_i)$$
$$+ \Delta(t_i \rightarrow t_{i+1}|z_i), \tag{3}$$

where $\gamma = (0, 1]$ averages $\Delta(t_i \rightarrow t_{i-1}|z_i)$ and $(z_{i-1} - z_i)$ for the time-slot $[t_{i-1}, t_i]$. The minus sign in front of the two quantities are due to the reverse integration direction. The quantity $-(z_{i-1} - z_i)$ is the previously computed integration approximation for $\int_{t_{i-1}}^{t_i} d(z, t)dt$. We note that negative $\gamma$ values are not applicable to the diffusion models considered in (Zhang et al., 2023a).

**On reversibility of BDIA**: The update expression (3) is carefully designed in (Zhang et al., 2023a) to enable diffusion

inversion for round-trip image editing. By reformulating (3), $z_{i-1}$ can be easily computed in terms of $(z_i, z_{i+1})$. We note that due to the nature of the floating-point datatype, there might be error accumulation in round-trip image reconstruction, where the corresponding diffusion states in the forward and reverse process are not identical. In practice, error accumulation of BDIA in diffusion inversion is not a big issue (Zhang et al., 2023a) due to the fact that the number of timesteps is generally set to be small (e.g., 50 timesteps in either forward or reverse process) to make the time complexity reasonable.

One main difference between diffusion inversion and reversible transformers is that no gradient needs to be back-propagated in diffusion inversion for updating the DNN model. As a result, even if there is error accumulation in diffusion inversion, it is less severe than in reversible transformers where error accumulation in online back-propagation would slow down the training convergence or even make the training fail especially for very deep models like LLMs. In next section, we will explain how to design exact bit-level reversible transformers in the training process to avoid any error-accumulation while at the same time, maintaining the architectures of the transformer in the inference procedure.

## 4. Exact Bit-Level Reversible Transformers via BDIA

In this section, we first briefly review the transformer update expressions from the ODE viewpoint. We then propose the BDIA-transformer that enables exact bit-level reversibility with activation quantization. Specially, we will demonstrate how each of the two $\gamma$ values $\{-0.5, 0.5\}$ averages consecutive integration approximations. The training of a BDIA-transformer can then be interpreted as employing different ODE solvers for different training samples in a random manner. In addition, we explain why additional lightweight side-information is required to be stored to account for the binary quantization loss in online back-propagation.

### 4.1. Revisiting transformer update expression

A typical transformer block consists of the attention function (denoted as $f(\cdot)$) and the function of feed-forward network (FFN) (denoted as $g(\cdot)$), of which the trainable parameters are generally different for different block indices. Accordingly, the output $x_{k+1}$ of the $k$th transformer block can be mathematically represented in terms of the input $x_k$ as

$$x_{k+1} = x_k + \underbrace{f_k(x_k) + g_k(x_k + f_k(x_k))}_{h_k(x_k)}, \tag{4}$$

where we use $h_k(x_k)$ to denote the overall residual quantity that includes both the attention and FFN functions. For simplicity, we omit the pre-normalisation operations

in (4), since these in fact do not affect the design of BDIA-transformers later on.

It is well known from the literature (Chen et al., 2018) that the $k$th forward step in (4) can be roughly viewed as the Euler integration approximation of an ODE at timestep $t_k$:

$$\boldsymbol{x}_{k+1} = \boldsymbol{x}_k + \boldsymbol{h}_k(\boldsymbol{x}_k) = \boldsymbol{x}_k + \overbrace{\tilde{\boldsymbol{d}}(\boldsymbol{x}_k, t_k)(t_{k+1} - t_k)}^{\Delta(t_k \rightarrow t_{k+1}|\boldsymbol{x}_k)} \quad (5)$$

$$\approx \boldsymbol{x}_k + \int_{t_k}^{t_{k+1}} \tilde{\boldsymbol{d}}(\boldsymbol{x}, t)dt,$$

where $\tilde{\boldsymbol{d}}(\boldsymbol{x}_k, t_k)$ denotes the gradient vector with $(\boldsymbol{x}_k, t_k)$ as the input. Both $\tilde{\boldsymbol{d}}(\boldsymbol{x}_k, t_k)$ and $(t_{k+1} - t_k)$ are implicitly learned via the composite function $\boldsymbol{h}_k(\boldsymbol{x}_k)$, which is alternatively denoted as $\Delta(t_k \rightarrow t_{k+1}|\boldsymbol{x}_k)$.

As explained later on, we will introduce BDIA into the update expression of (5). Note that (5) is a general update expression that not only includes the transformer but also ResNet (He et al., 2015) and its variants. It is noted in (Dupont et al., 2019) that ResNet can represent more functions than neural ODEs. Despite the less powerfulness of the ODE framework, we employ the framework to facilitate the design of reversible transformer and obtain promising empirical results.

### 4.2. BDIA-transformer without quantization

In this subsection, we first derive the update expressions of BDIA-transformer without quantization as an extension of (5), and then study the impact of the $\gamma$ values in $\{0.5, -0.5\}$. When $k = 0$, $\boldsymbol{x}_1$ can be computed by following (5) as

$$\boldsymbol{x}_1 = \boldsymbol{x}_0 + \boldsymbol{h}_0(\boldsymbol{x}_0) = \boldsymbol{x}_0 + \Delta(t_0 \rightarrow t_1|\boldsymbol{x}_0). \quad (6)$$

The update expression for $\boldsymbol{x}_{k+1}$ in the training process, $K - 1 \geq k \geq 1$, can be obtained by utilizing (2)-(3). Based on (5), we let

$$\Delta(t_k \rightarrow t_{k-1}|\boldsymbol{x}_k) = -\boldsymbol{h}_k(\boldsymbol{x}_k) \quad (7)$$
$$\Delta(t_k \rightarrow t_{k+1}|\boldsymbol{x}_k) = \boldsymbol{h}_k(\boldsymbol{x}_k). \quad (8)$$

By combining (7)-(8) and (2)-(3) with $i = k$, the update expression $\boldsymbol{x}_{k+1}$, $K - 1 \geq k \geq 1$, can be represented as

$$\boldsymbol{x}_{k+1} = \boldsymbol{x}_{k-1} + (1-\gamma)(\boldsymbol{x}_k - \boldsymbol{x}_{k-1}) + (1+\gamma)\boldsymbol{h}_k(\boldsymbol{x}_k) \quad (9)$$
$$= \gamma\boldsymbol{x}_{k-1} + (1-\gamma)\boldsymbol{x}_k + (1+\gamma)\boldsymbol{h}_k(\boldsymbol{x}_k), \quad (10)$$

where $\gamma$ is recommended to take values from $\{0.5, -0.5\}$ with equal probability per training sample per transform block, the impact of which will be explained in detail in the following. This is different from the work of BDIA-based diffusion inversion (Zhang et al., 2023a) where $\gamma$ has to be positive.

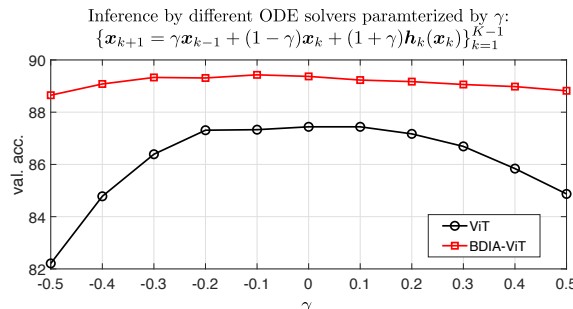

*Figure 1.* Validation performance of different ODE solvers parameterized by a single $\gamma$ parameter after training ViT and BDIA-ViT over CIFAR10. See Subsection 5.2 on how ViT and BDIA-ViT were trained. Each ODE solver in the inference procedure is realized by selecting $\gamma$ from $[-0.5, 0.5]$, which is fixed across all the transformer blocks for the same input image. The validation performance of BDIA-ViT is more robust than that of ViT.

In the inference stage, $\mathbb{E}(\gamma) = 0$ is taken to replace $\gamma$ in (10), which leads to a simpler update expression that only involves $(\boldsymbol{x}_k, \boldsymbol{x}_{k+1})$:

$$\boldsymbol{x}_{k+1} = \boldsymbol{x}_k + \boldsymbol{h}_k(\boldsymbol{x}_k), \quad (11)$$

which is identical to the original transformer update expression (4)-(5).

The above analysis suggests that the BDIA training technique is a good candidate for fine-tuning transformer-based models because BDIA retains the standard architecture in the inference procedure. Fine-tuning a pre-trained model is typically performed on a small or intermediate-sized dataset, which is prone to overfitting. Subsection 5.1 and Appendix E demonstrate successfully applying BDIA to fine-tune GPT2 medium using a small dataset.

**Impact of $\gamma$ parameter**: We now study the impact of the $\gamma$ parameter in (10). When $\gamma = 0.5$, it follows from (2)-(3) and (7)-(8) that the two integrations $\int_{t_{k-1}}^{t_k} \tilde{\boldsymbol{d}}(\boldsymbol{x}_\tau, \tau)d\tau$ and $\int_{t_k}^{t_{k+1}} \tilde{\boldsymbol{d}}(\boldsymbol{x}_\tau, \tau)d\tau$ of the transformer ODE are approximated as

$$\int_{t_{k-1}}^{t_k} \tilde{\boldsymbol{d}}(\boldsymbol{x}_\tau, \tau)d\tau \approx 0.5(\boldsymbol{x}_k - \boldsymbol{x}_{k-1}) + 0.5\boldsymbol{h}_k(\boldsymbol{x}_k) \quad (12)$$

$$\int_{t_k}^{t_{k+1}} \tilde{\boldsymbol{d}}(\boldsymbol{x}_\tau, \tau)d\tau \approx \boldsymbol{h}_k(\boldsymbol{x}_k), \quad (13)$$

where the integration over $[t_{k-1}, t_k]$ is computed as the weighted average of two consecutive integration approximations: $(\boldsymbol{x}_k - \boldsymbol{x}_{k-1})$ and $\boldsymbol{h}_k(\boldsymbol{x}_k)$.

When $\gamma = -0.5$, the two integrations $\int_{t_{k-1}}^{t_k} \tilde{\boldsymbol{d}}(\boldsymbol{x}_\tau, \tau)d\tau$ and

$\int_{t_k}^{t_{k+1}} \tilde{\boldsymbol{d}}(\boldsymbol{x}_\tau, \tau)d\tau$ are approximated differently, given by

$$\int_{t_{k-1}}^{t_k} \tilde{\boldsymbol{d}}(\boldsymbol{x}_\tau, \tau)d\tau \approx (\boldsymbol{x}_k - \boldsymbol{x}_{k-1}) \tag{14}$$

$$\int_{t_k}^{t_{k+1}} \tilde{\boldsymbol{d}}(\boldsymbol{x}_\tau, \tau)d\tau \approx 0.5\boldsymbol{h}_k(\boldsymbol{x}_k) + 0.5(\boldsymbol{x}_k - \boldsymbol{x}_{k-1}). \tag{15}$$

In this case, the integration over $[t_k, t_{k+1}]$ is computed by averaging $(\boldsymbol{x}_k - \boldsymbol{x}_{k-1})$ and $\boldsymbol{h}_k(\boldsymbol{x}_k)$.

**Training via an ensemble of ODE solvers**: We use $\gamma_k$ to denote the random variable for the $k$th transformer block taking values in $\{0.5, -0.5\}$ with equal probability. The above analysis of (12)-(15) implies that each training sample goes through a particular ODE solver determined by sampling a set of $K-1$ random variables $\{\gamma_k\}_{k=1}^{K-1}$, which specifies the integration path from the bottom transformer block until the top one. Due to randomness, different training samples will go through different ODE solvers. There are in total $2^{K-1}$ different ODE solvers, where each one corresponds to a unique integration path across the $K$ transformer blocks. Intuitively speaking, if we assume that each of the individual ODE solvers is well behaved (i.e., its training loss decays to a small value), then a convex combination will also converge to a small value.

In principle, different ODE solvers can also be applied in the inference procedure. For instance, one can set $\gamma$ to be constant within $[-0.5, 0.5]$ in (10) across all the transformer blocks for the same input. Fig. 1 demonstrates the validation performance of those different ODE solvers in the inference procedure for both trained BDIA-ViT and ViT over CIFAR10. It is clear that the validation performance of BDIA-ViT is much more insensitive to the single $\gamma$ parameter than that of ViT. The reason for the sensitivity of ViT to $\gamma$ in Fig. 1 is because only a single ODE solver is trained for ViT.

**On similarity to dropout technique**: From a high level point of view, the training of BDIA-transformer is similar to the conventional dropout technique (Srivastava et al., 2014) to a certain extent. Dropout essentially attempts to train an ensemble of subnetworks of the entire DNN model when minimizing the objective function while BDIA-transformer intends to train an ensemble of different ODE solvers. In the inference procedure, an average of the ensemble of subnetworks in dropout is utilized while in BDIA-transformer, the expectation $\mathbb{E}[\gamma_k] = 0$ is employed to replace $\gamma_k$. In the experiments we investigated the joint impact of dropout and BDIA for the image classification task, and obtained positive results. See Table 5 for details.

*Remark* 4.1. If reversibility is not of concern, one can freely specify the values for the random variables $\{\gamma_k\}_{k=1}^{K-1}$ in the training process for performance improvement as long as its distribution is symmetric around 0 such that $\mathbb{E}[\gamma_k] = 0$ for

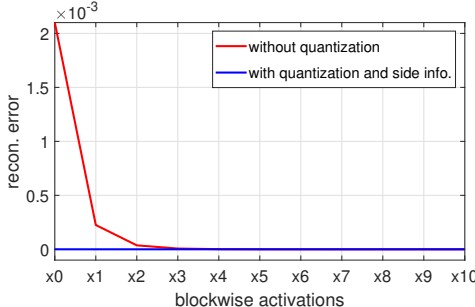

*Figure 2.* Demonstration of the accumulated reconstruction errors for two scenarios when training BDIA-GPT2 with 12 transformer blocks using the setup $\gamma_k \in \{0.5, -0.5\}$, $k = 1, \ldots, 11$. The two scenarios are BDIA without quantization by following (16), and BDIA with quantization and side information by following (24). See also Fig. 6 for visualization of the reconstruction errors.

maintaining the original DNN architectures in the inference procedure. See Table 3 for ablation study of the impact of the $\{\gamma_k\}_{k=1}^{K-1}$ parameter on the validation performance when training BDIA-transformer for image classification.

### 4.3. On exact bit-level reversibility of BDIA-transformer with quantization

As explained in Section 1, one strategy for reducing memory consumption in training transformer models like LLMs is to perform online back-propagation. That is, the intermediate activation outputs from the top transformer block until the bottom one are computed online when performing back-propagation to update the model. In this subsection, we first discuss the reversibility issue of the update expression (10). We then consider performing activation quantization to enable exact bit-level reversibility. We note that lightweight side information is required to be stored per transformer block for lossless online back-propagation.

**Limitation of the reversibility of (10)**: The update expression (10) is only theoretically reversible. That is, $\boldsymbol{x}_{k-1}$ can be computed in terms of $(\boldsymbol{x}_k, \boldsymbol{x}_{k+1})$ as

$$\boldsymbol{x}_{k-1} = \frac{1}{\gamma_k}\boldsymbol{x}_{k+1} - \frac{1 - \gamma_k}{\gamma_k}\boldsymbol{x}_k - \frac{1 + \gamma_k}{\gamma_k}\boldsymbol{h}_k(\boldsymbol{x}_k), \tag{16}$$

where we use $\gamma_k$ to indicate that different transformer blocks have their respective random variables. In practice, the setup $\gamma_k \in \{0.5, -0.5\}$, $k = 1, \ldots, K-1$, would lead to non-negligible error accumulation especially for very deep transformer models. The factor $\frac{1}{\gamma_k} = \pm 2$ in front of $\boldsymbol{x}_{k+1}$ would amplify the error when $k$ decreases from $K-1$ to 1, making the online back-propagation unstable. Fig. 2 illustrates that the reconstruction error without quantization increases significantly when applying the online-back-propagation from the last transformer block back to the first.

**BDIA-transformer with quantization**: To allow for lossless online back-propagation, we propose to perform activation quantization. In particular, we use $\mathcal{Q}_l[\cdot]$ to denote

quantization to the bit-level precision of $2^{-l}$, given by

$$Q_l[y] = \text{round}[y/2^{-l}]2^{-l}. \quad (17)$$

Upon introducing $Q_l[\cdot]$, the new update expression for BDIA-transformer can be represented as

$$x_0 \leftarrow Q_l[x_0], \quad (18)$$

$$x_1 = x_0 + Q_l[h_0(x_0)] \quad (19)$$

$$s_{k-1}[m] = \begin{cases} 1 & \text{if } \text{mod}(x_{k-1}[m]/2^{-l}, 2) = 1 \\ 0 & \text{otherwise} \end{cases} \quad k \geq 1 \quad (20)$$

$$x_{k+1} = Q_l[\gamma_k(x_{k-1} + s_{k-1}2^{-l})] + Q_l[(1-\gamma_k)x_k + (1+\gamma_k)h_k(x_k)] \quad k \geq 1, \quad (21)$$

where $\gamma_k \in \{0.5, -0.5\}$, and $x_{k-1}[m]$ denotes the $m$th element of $x_{k-1}$. The $m$th element $s_{k-1}[m]$ indicates if the integer value $x_{k-1}[m]/2^{-l}$ is odd or not.

**Lemma 4.2.** *Suppose $\{x_k\}_{k=0}^K$ are computed sequentially by following (18)-(21). Then the intermediate activation output $x_k$ has fixed-point precision of $2^{-l}$:*

$$x_k = Q_l[x_k] \qquad K \geq k \geq 0. \quad (22)$$

We refer to the binary vector $s_{k-1}$ in (20) as the lightweight side information computed based on $x_{k-1}$. The vector $s_{k-1}$ essentially captures the 1-bit quantization loss of $Q_l[\gamma_k x_{k-1}]$ per element, and is crucial for recovering $x_{k-1}$ exactly from $(x_k, x_{k+1})$. We now characterize the first quantization operation on the right hand side (RHS) of (21) with the following proposition:

**Proposition 4.3.** *Suppose the conditions in Lemma 4.2 hold. Then for any $k \geq 1$, we have*

$$Q_l[\gamma_k(x_{k-1} + s_{k-1}2^{-l})] = \gamma_k(x_{k-1} + s_{k-1}2^{-l}). \quad (23)$$

The proof for Proposition 4.3 can be found in Appendix B. Equ. (23) indicates that the quantization operation has no effect on $\gamma_k(x_{k-1} + s_{k-1}2^{-l})$, which naturally facilitates the exact reconstruction of $x_{k-1}$ from $(x_k, x_{k+1})$.

**On reversibility of (21) by storing lightweight side information**: Suppose in each forward pass in the training process, all the side information $\{s_{k-1}\}_{k=1}^{k=K-1}$ is stored in the memory. Then exact bit-level reversibility is guaranteed:

**Proposition 4.4.** *Suppose the conditions in Lemma 4.2 hold and the side information $\{s_{k-1}\}_{k=1}^{k=K-1}$ is available. Then $\{x_{k-1}\}_{k=1}^{K-1}$ can be reconstructed exactly in the reverse order with the initial state-pair $(x_{K-1}, x_K)$:*

$$x_{k-1} = \frac{1}{\gamma_k}x_{k+1} - s_{k-1}2^{-l}$$
$$- \frac{1}{\gamma_k}Q_l[(1-\gamma_k)x_k + (1+\gamma_k)h_k(x_k)], \quad (24)$$

*where $k = K-1, \ldots, 1$.*

See Appendix C for the proof. The lossless reconstruction (24) ensures that the computed gradients in the online back-propagation would not deviate from those in conventional back-propagation, which is desirable in very deep transformer models. In Fig. 2, the blue line shows the exact reconstruction of the intermediate activation values, as enabled by (24) during online back-propagation.

Finally we briefly consider the inference procedure. Again we replace $\gamma_k$ in (21) by $\mathbb{E}(\gamma_k) = 0$. As a result, the update expression (21) can be simplified to be

$$x_{k+1} = Q_l[x_k + h_k(x_k)] \quad k \geq 1. \quad (25)$$

The only difference of (25) w.r.t. the original transformer update expression (4) is that the quantization operation $Q_l[\cdot]$ is performed for each activation output.

*Remark 4.5.* The particular choice of $\gamma_k \in \{0.5, -0.5\}$ is also motivated by exact bit-level reversibility. When $\gamma$ is a power of two such as $0.5 = 2^{-1}$, the result of a product $\gamma \cdot x$ is equivalent to shifting the mantissa of $x$ by one bit or decrementing its exponent, so at most exactly one bit is lost. This property is not true for arbitrary (non-power-of-2) values of $\gamma$.

**Limitations of BDIA-transformer**: To our best knowledge, all existing reversible DNN models do not need to store any lightweight side information in the forward process. The primary objective of most existing works is to design reversible DNN models (e.g., RevViT) that produce comparable validation performance as the original counterparts. From the perspective of memory reduction, existing reversible DNN models are more efficient than BDIA-transformer.

On the other hand, BDIA-transformer is designed to not only save training memory but also improve the generalization performance via model regularization. Our method is the first of its kind that attempts to maintain the transformer architecture in the inference procedure.

## 5. Experiments

We evaluated BDIA-transformer for four different tasks: (1) fine-tuning GPT2 using the BDIA training technique for natural language generation (NLG); (2) training BDIA-ViT for image classification; (3) training BDIA-transformer for language translation; (4) training BDIA-GPT2 for text generation. The hyper-parameter $l$ for quantization in all four tasks was set to $l = 9$. Four open-source repositories were used in the experiments (see Table 7 in Appendix G). Due to the space constraint, we put the experimental results for the fourth task in Appendix F.

In brief, the obtained results from the first three tasks indicate that the BDIA technique significantly improves the validation performance of the original counterparts due to

*Table 1.* Performance for fine-tunning GPT-2 medium (M) on the E2E NLG Challenge. For all metrics, higher is better.

| Method | trainable param. | BLEU | NIST | MET | ROUGE-L | CIDEr |
|--------|------------------|------|------|-----|---------|-------|
| LoRA | 0.35M | **69.0** | **8.70** | 46.4 | 71.3 | **2.51** |
| FT | 354.92M | 66.8 | 8.52 | 46.3 | 70.3 | 2.37 |
| FT-BDIA | 354.92M | 68.6 | 8.65 | **46.6** | **71.3** | 2.49 |

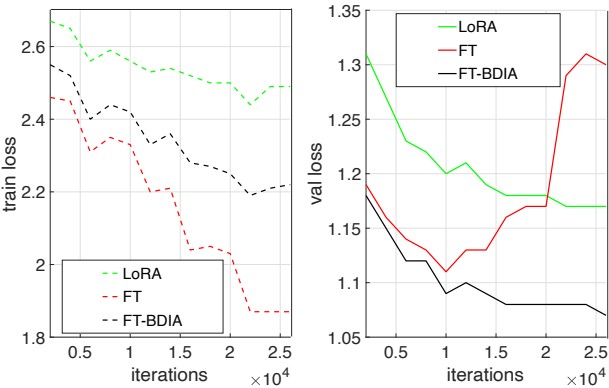

*Figure 3.* Performance comparison when fine-tuning GPT2 Medium for the E2E challenge. $\{\gamma_k\}_{k=1}^{K-1}$ in the BDIA training procedure were randomly drawn from $\{\pm 0.5\}$ per training sample. FT-BDIA refers to "fine-tuning via BDIA training technique" while FT refers to "fine-tuning directly".

the model regularization effect of the random $\{\gamma_k\}_{k=1}^{K-1}$ variables. The results for the 4th task demonstrate that BDIA-GPT2 alleviates the overfitting issue of GPT2 significantly for limited training data. RevViT from (Mangalam et al., 2023) was also tested for the second task. It was found that RevViT does not always improve the performance of ViT.

## 5.1. On fine-tuning GPT2 medium (M) for NLG

In the first experiment, we consider fully fine-tuning GPT2 medium (354.92M parameters in total) for the natural language generation (NLG) task by using the E2E dataset (Novikova et al., 2017). We adopt the open-source for GPT2 (LoRA) in (Hu et al., 2021) (the second github link of Table 7) to evaluate the BDIA training technique. For comparison, we also fine-tune GPT2 directly and via the LoRA technique with the default setup of $(\text{rank}, \alpha) = (4, 32)$. It is worth noting that the LoRA technique only needs to train a very small number of parameters.

It is clear from Fig. 3 that the BDIA training technique leads to the best validation performance in the end of training while direct fine-tuning of GPT2 exhibits the over-fitting issue. It is plausible that the validation losses for LoRA are larger than those for FT-BDIA because LoRA only trains a small number of parameters.

Table 1 summarizes results for 5 metrics. It is seen from

the table that results for FT-BDIA are consistently better than those for FT. On the other hand, even though the LoRA technique is light-weight, its performance is promising.

*Remark* 5.1. We have also investigated the joint impact of LoRA and BDIA when fine-tuning GPT2 M for NLG, and again we obtained positive results. See Appendix E for details. In brief, BDIA helps with improving the validation performance for LoRA based fine-tuning for the considered task. To enhance the impact of BDIA in the fine-tuning process, it is recommended to set a high value for the $\alpha$ parameter in LoRA.

## 5.2. On training BDIA-ViT

In this experiment, we trained BDIA-ViT with $K=6$ transformer blocks on CIFAR10 and CIFAR100 by using a single 2080 Ti GPU. The performance of ViT and RevViT (Mangalam et al., 2023) was also evaluated to facilitate comparison. When implementing BDIA-ViT, the $\{\gamma_k\}_{k=1}^{K-1}$ parameters were drawn from $\{-0.5, 0.5\}$ with equal probability per training sample. In addition, we utilized the SET-Adam optimizer (Zhang, 2024) in the training process with the configuration $(\eta_0, \beta_1, \beta_2, \epsilon) = (1e-4, 0.9, 0.999, 1e-18)$, where $\eta_0$ denotes the initial learning rate. The dropout rate was set to 0.1 to reduce over-fitting. The remaining training setups follow directly from the original open source (i.e., the first github link of Table 7). Three experimental repetitions were performed for each training setup to mitigate the effect of the random seed.

**Performance comparison**: Table 2 summarizes the obtained validation accuracy, the peak memory usages, and the average training time per epoch. It is clear that BDIA-ViT produces significantly higher validation accuracy than ViT and RevViT for both CIFAR10 and CIFAR100. Fig. 4 further visualizes the training and validation curves. It is seen that even though the training loss of BDIA-ViT is higher than the other two models across all the epochs, the validation accuracy improves remarkably in the end of training. This indicates that training an ensemble of ODE solvers parameterized (see Subsection 4.2) by $\{\gamma_k\}_{k=1}^{K-1}$ indeed regularizes BDIA-transformer properly.

In contrast, RevViT yields either inferior or comparable validation performance to ViT (Table 2). This may be because RevViT modifies the architectures of ViT considerably to enable reversibility and therefore implicitly imposes uncontrolled regularization on the original transformer model. On the other hand, the regularisation introduced in our technique is motivated by averaging consecutive integration approximations of an ODE in the original transformer, which leads to consistent performance gain across different datasets.

It is also clear from Table 2 that RevViT is most memory-efficient. BDIA-ViT improves on the validation perfor-

*Table 2.* Validation accuracy (in percentage), peak memory consumption, and average training time per epoch for training three models over CIFAR10 and CIFAR100. $\{\gamma_k\}_{k=1}^{K-1}$ in the BDIA training procedure were drawn from $\{\pm 0.5\}$ per training sample. The peak memory includes both the model parameters and the training states for a batchsize of 128. The dropout rate was set to 0.1. See also Fig. 7 for training curves on CIFAR-10 plotted against wall-clock time.

| | RevViT (Mangalam et al., 2023) | | | ViT | | | BDIA-ViT | | |
|---|---|---|---|---|---|---|---|---|---|
| | val. acc. | peak memory | train. time per epoch | val. acc. | peak memory | train. time per epoch | val. acc. | peak memory | train. time per epoch |
| CIFAR10 | 86.22±0.42 | 572.7MB | 41.8 s | 88.15±0.55 | 1570.6MB | 30.6 s | **89.10**±0.38 | 693.4MB | 47.0 s |
| CIFAR100 | 61.89±0.31 | 572.7MB | 41.8 s | 61.86±0.47 | 1570.6MB | 30.6 s | **66.09**±0.80 | 693.4MB | 47.0 s |

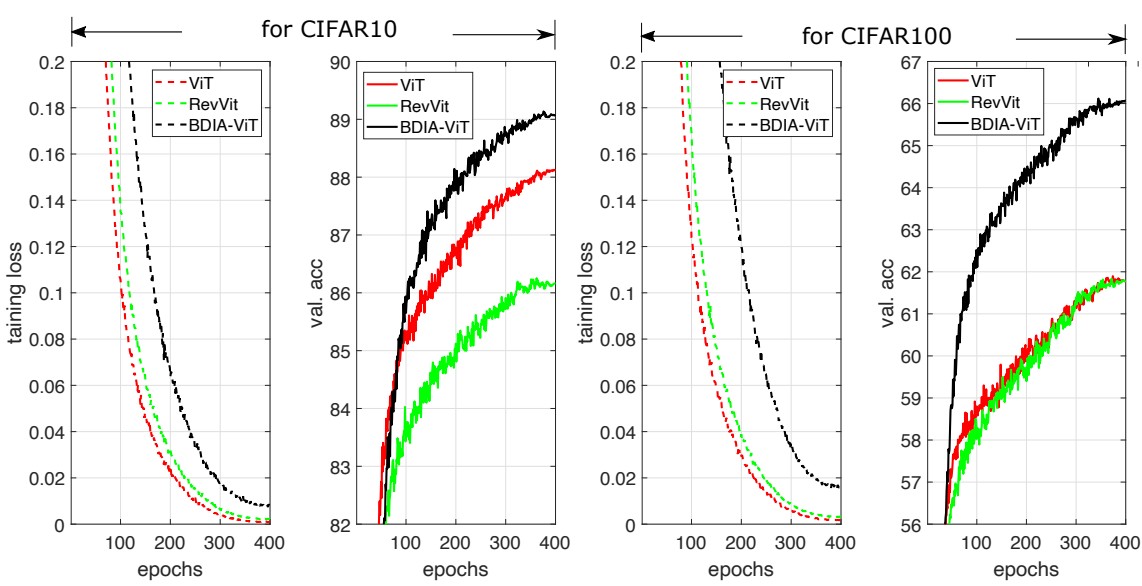

*Figure 4.* Performance comparison of ViT, RevViT (Mangalam et al., 2023), and BDIA-ViT for image classification over CIFAR10 and CIFAR100. $\{\gamma_k\}_{k=1}^{K-1}$ in the training procedure of BDIA-ViT were drawn from $\{\pm 0.5\}$ per training sample. The dropout rate was 0.1.

mance of RevViT at the cost of slightly more memory, needed to store the lightweight side information $\{s_k\}_{k=0}^{3}$ of the first 4 transformer blocks. We can also conclude from the table that online back-propagation does indeed significantly reduce the training memory.

Lastly, we briefly discuss the average training time per epoch. It is clear from the table that both ReViT and BDIA-ViT require additional training time for recomputation of the intermediate activation values in online-backpropagation. BDIA-ViT needs slightly more time than ReViT because the method needs to handle the side information, the binary random variables, and quantization. See Fig. 7 for training curves on CIFAR-10 plotted against wall-clock time.

**Ablation study regarding the $\{\gamma_k\}_{k=1}^{K-1}$ parameters**: As we mentioned in Remark 4.1, if reversibility is not of primary concern, different values for $\{\gamma_k\}_{k=1}^{K-1}$ can be employed for model regularisation. We conducted ablation study by evaluating the impact of $\gamma_k \in \{0.0, \pm 0.25, \pm 0.5, \pm 0.6\}$, $k = 1, \ldots, K-1$, on the validation performance of BDIA-transformer for CIFAR10. For doing so, both the quantization and online back-propagation

operations were turned off in BDIA-transformer. We emphasize that the non-zero $\gamma_k$ values were only utilized in the training process. At the inference stage, $\mathbb{E}[\gamma_k] = 0$ was used for computing the validation accuracy, which in fact reduces to the standard transformer architecture.

*Table 3.* Impact of the $\{\gamma_k\}_{k=1}^{K-1}$ parameter in BDIA-ViT (w.o. quantization and w.o. online back-propagation) on the validation accuracy in percentage over CIFAR10. The dropout rate was 0.1.

| $\{\gamma_k\}_{k=1}^{K-1}$ | 0.0 | $\{\pm 0.25\}$ | $\{\pm 0.5\}$ | $\{\pm 0.6\}$ |
|---|---|---|---|---|
| CIFAR10 | 88.15±0.55 | 88.79±0.29 | **89.12**±0.22 | 88.89±0.15 |

Table 2 summarizes the obtained validation accuracy for different setups of the $\{\gamma_k\}_{k=1}^{K-1}$ parameters. It is clear that when $\{|\gamma_k| > 0\}_{k=1}^{K-1}$, the performance of BDIA-transformer improves considerably in comparison to that of the conventional ViT (i.e., corresponding to BDIA-ViT with $\{\gamma_k = 0.0\}_{k=1}^{K-1}$). The setup of $\{\gamma_k = \pm 0.5\}_{k=1}^{K-1}$ performs the best. In general, the larger the magnitude of the $\{\gamma_k\}_{k=1}^{K-1}$ parameters in $[0, 0.6]$, the slower the training speed (see Fig. 4). In practice, one can tune the magnitude of the $\{\gamma_k\}_{k=1}^{K-1}$ parameters within $[0, 0.6]$ to trade-off between

the training speed and the validation performance.

**Ablation study regarding the quantization level**: We have also investigated the impact of the quantization levels of $\{5, 0, -2\}$ on the performance of both ViT and BDIA-ViT over CIFAR10, where $l = -2$ corresponds to a very coarse quantization operation. The $\{\gamma_k\}_{k=1}^{K-1}$ variables in BDIA-ViT were drawn randomly from $\{\pm 0.5\}$ per training sample.

*Table 4.* Impact of the quantization level in ViT and BDIA-ViT on the validation accuracy in percentage over CIFAR10.

| quantization level $l$ | 5 | 0 | -2 |
|---|---|---|---|
| ViT | 87.90 | 87.76 | **84.31** |
| BDIA-ViT | **89.50** | **88,.47** | 84.16 |

As shown in Table 4, when the quantization level $l$ drops from 5 to -2, the performance of ViT and BDIA-ViT decreases as expected. One can also observe from the table that when $l \in \{5, 0\}$, BDIA-ViT performs consistently and significantly better than ViT. When $l = -2$, there is a large performance drop in both methods due to the very coarse quantization effect. For that case, ViT performs slightly better than BDIA-ViT. This likely is because the very coarse quantization dominates the performance and negatively affects the BDIA-training technique.

**Ablation study regarding the dropout rates**: As we discussed earlier, the dropout technique trains an ensemble of subnetworks, while BDIA trains an ensemble of ODE solvers. It is of great interest to investigate the joint impact of dropout and BDIA. We conducted additional experiments by setting the dropout rate to 0.0 and 0.2 when testing BDIA-ViT. Table 5 shows the obtained validation accuracies, where the results for dropout rate of 0.1 are from Table 2.

*Table 5.* Impact of dropout rates in ViT and BDIA-ViT on the validation accuracy percentage. In BDIA training $\{\gamma_k\}_{k=1}^{K-1}$ were randomly drawn from $\{\pm 0.5\}$ per training sample.

| dropout rate | 0.0 | | 0.1 | | 0.2 | |
|---|---|---|---|---|---|---|
| | ViT | BDIA-ViT | ViT | BDIA-ViT | ViT | BDIA-ViT |
| CIFAR10 | 86.27 | **89.20** | 88.15 | **89.10** | 87.24 | **88.22** |
| CIFAR100 | 59.13 | **64.45** | 61.86 | **66.09** | 61.68 | **64.24** |

It is seen from Table 5 that for different dropout rates and different datasets, the BDIA training technique improves the validation performance consistently and considerably. In brief, the dropout and BDIA are complimentary regularization techniques when training transformers.

### 5.3. On training BDIA-transformer for English-French translation

Language translation is a classical natural language processing (NLP) task where the transformer shows a large performance gain over other DNN models. We adopted an

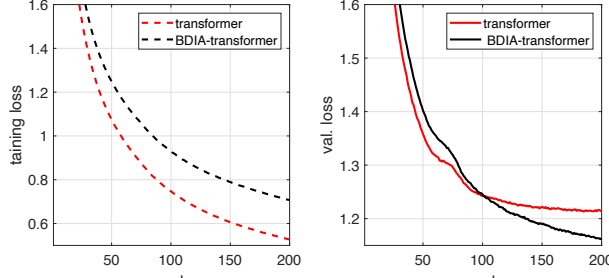

*Figure 5.* Performance comparison for English to French translation. $\{\gamma_k\}_{k=1}^{K-1}$ in the BDIA training procedure were randomly drawn from $\{\pm 0.5\}$ per training sample.

existing open-source repository (i.e., the third github link of Table 7) in our experiment. The dataset being used is from Kaggle (Kelly, 2020). The tested BDIA-transformer has six transformer blocks in both the encoder and decoder, respectively. The BDIA update expressions were implemented in both the encoder and decoder. The performance of the conventional transformer was tested as a reference.

Fig. 5 visualizes the training and validation losses of the two tested models. It is clear that the obtained curves in Fig. 5 exhibit similar properties to those in Fig. 4. That is, even though the training loss of BDIA-transformer is higher than the conventional transformer across all the epochs, its validation loss is significantly lower after epoch 200.

## 6. Conclusions

We have proposed the BDIA training algorithm for transformers. Firstly, each transformer block is taken as the Euler integration approximation for solving an ODE. The BDIA technique is then applied to average every two consecutive integration approximations in a transformer as a regularizer via a set of random variables $\{\gamma_k\}_{k=1}^{K-1}$, one variable per transformer block. The training process can then be understood as training an ensemble of ODE solvers parameterized by $\{\gamma_k\}_{k=1}^{K-1}$. Exact reversibility for lossless online back-propagation is achieved by activation quantization and storing only lightweight side information. In the inference procedure, the variables $\gamma_k$ are replaced with their expectation $\mathbb{E}[\gamma_k] = 0$, which reduces the update expression to that of a conventional transformer up to activation quantization.

Experiments on natural language generation, translation, and image classification show that BDIA-transformer produces significantly better validation performance than the corresponding baseline transformers while, at the same time, reducing training memory by performing online back-propagation. In comparison, the experiments with RevVit demonstrate that it yields either inferior or comparable validation performance to that of ViT. BDIA-transformer is also attractive for fine-tuning since it requires minimal architecture changes, needs very little extra memory, and is complementary to LoRA.

## Impact Statement

Transformer architectures are widely used in various contexts including, for example, LLMs for natural language processing and diffusion models for image and video processing. Fine-tuning these models with either small or intermediate data sizes for diverse downstream tasks is a common practice. Our BDIA training technique offers an attractive approach for fine-tuning. This is because BDIA trains an ensemble of ODE solvers to reduce over-fitting and lower training memory requirements while simultaneously retaining the original transformer architecture in the inference procedure.

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

## A. Demonstration of Reversibility Issue of BDIA Without Quantization on a Log Scale

We note that the reconstruction errors in Fig. 2 are plotted on a linear-scale. To better visualize the results, we include Fig. 6 for the case of BDIA without quantization by following (16). The errors are plotted on a log scale.

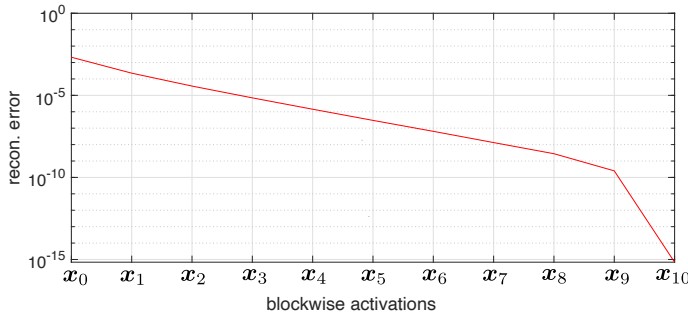

*Figure 6.* Demonstration of the accumulated reconstruction error in log-scale by following (16) with the setup $\gamma_k \in \{0.5, -0.5\}$, $k = 1, \ldots, K-1$, when training BDIA-GPT2 with 12 transformer blocks.

## B. Proof of Proposition 4.3

*Proof.* Equation (23) in Proposition 4.3 can be proved by simple algebra. It is known from Lemma 4.2 that $\mathcal{Q}_l[\boldsymbol{x}_{k-1}] = \boldsymbol{x}_{k-1}$ holds for any $k \geq 1$. For each element of the expression on the left hand side (LHS) of (23), we have

$$
\begin{aligned}
&\mathcal{Q}_l[\gamma_k(\boldsymbol{x}_{k-1}[m] + \boldsymbol{s}_{k-1}[m]2^{-l})] \\
&= \text{round}\Big[\gamma_k(\boldsymbol{x}_{k-1}[m]/2^{-l} + \boldsymbol{s}_{k-1}[m])\Big]2^{-l}.
\end{aligned}
\tag{26}
$$

By using (17) and the definition of $\boldsymbol{s}_{k-1}[m]$ from (20), it is immediate that the scalar $(\boldsymbol{x}_{k-1}[m]/2^{-l} + \boldsymbol{s}_{k-1}[m])$ in (26) is an even integer number. Since $\gamma_k \in \{-0.5, 0.5\}$, the round operation in (26) has no real effect and can be removed. Therefore, (26) can be rewritten as

$$
\begin{aligned}
&\mathcal{Q}_l[\gamma_k(\boldsymbol{x}_{k-1}[m] + \boldsymbol{s}_{k-1}[m]2^{-l})] \\
&= \Big[\gamma_k(\boldsymbol{x}_{k-1}[m]/2^{-l} + \boldsymbol{s}_{k-1}[m])\Big]2^{-l} \\
&= \gamma_k(\boldsymbol{x}_{k-1}[m] + \boldsymbol{s}_{k-1}[m]2^{-l}).
\end{aligned}
\tag{27}
$$

The proof is complete. $\qquad\square$

## C. Proof of Proposition 4.4

*Proof.* Also (24) in Proposition 4.4 can be proved by simple algebra. Firstly, plugging (23) into (21) produces

$$
\boldsymbol{x}_{k+1} = \gamma_k \underbrace{(\boldsymbol{x}_{k-1} + \boldsymbol{s}_{k-1}2^{-l})}_{\text{2nd term}} + \underbrace{\mathcal{Q}_l[(1-\gamma_k)\boldsymbol{x}_k + (1+\gamma_k)\boldsymbol{h}_k(\boldsymbol{x}_k)]}_{\text{3rd term}}.
\tag{28}
$$

where $\underbrace{\boldsymbol{x}_{k+1}}_{\text{1st term}}$

It is known from Lemma 4.2 and (20) that all three terms in (28) have a fixed-point precision of $2^{-l}$. As a result, the equality in (28) holds without any error. Multiplying with $\frac{1}{\gamma_k} = \pm 2$ on both sides of (28) and rearranging the quantities produces

$$
\boldsymbol{x}_{k-1} = \frac{1}{\gamma_k}\boldsymbol{x}_{k+1} - \boldsymbol{s}_{k-1}2^{-l} - \frac{1}{\gamma_k}\mathcal{Q}_l[(1-\gamma_k)\boldsymbol{x}_k + (1+\gamma_k)\boldsymbol{h}_k(\boldsymbol{x}_k)],
\tag{29}
$$

which again holds without any error. The proof is complete. $\qquad\square$

## D. Training Time Comparison

This Appendix discusses training effort in more detail. Table 2 includes the average training time per epoch for CIFAR10 and CIFAR100 for three algorithms and that Fig. 4 shows the training and validation curves over 400 epochs. Fig. 7 in this Appendix shows the training curves for CIFAR10 against wall-clock time for 400 epochs.

It is seen from Fig. 7 that BDIA-ViT with online backpropagation needs slightly more time to cover 400 epochs than RevViT with online backpropagation. This is because BDIA-ViT needs to handle the lightweight side information, binary random variables, and quantization. ViT is most time efficient as it does not need to recompute the intermediate activation values as the other two methods.

While the final training loss of BDIA-ViT is higher, as indicated in Fig. 7, its validation accuracy is better than those of the other two methods (see Fig. 4). The above property can be explained by the fact that BDIA trains an ensemble of ViT models using different ODE solvers and thus can be viewed as a complimentary regularization technique to dropout.

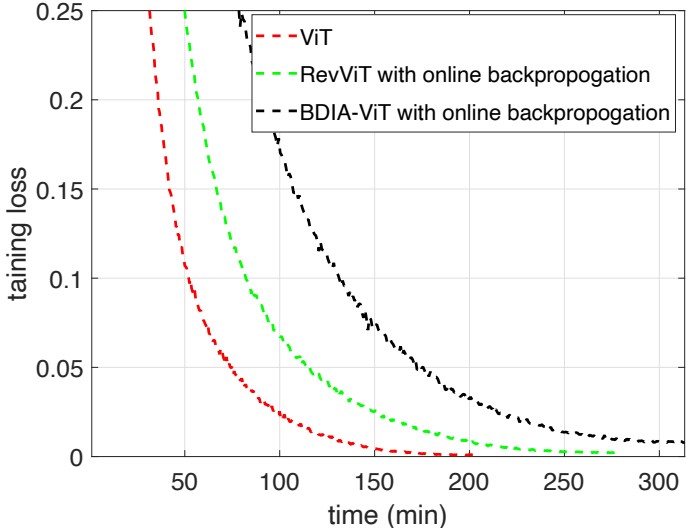

*Figure 7.* Training time comparison of ViT, RevViT with online backpropagation (Mangalam et al., 2023), and BDIA-ViT with online backpropagation for image classification over CIFAR10. Each training curve corresponds to 400 training epochs. Note that the BDIA technique can outperform in validation accuracy even when the training loss is higher, as seen in Fig. 4.

## E. Evaluation of Joint Impact of LoRA and BDIA

It has become common practice to adopt the LoRA adapter when fine-tuning an LLM with either small or intermediate data sizes. One major advantage of using LoRA is that it accelerates the training process with limited computational resources as it requires only a relatively small number of parameters to be trained. Since the original LLM model parameters are kept frozen throughout the process, the knowledge learned in the pre-training is less likely to be forgotten. On the other hand, it has been reported in the literature that, even with the LoRA approach, overfitting can still occur (Lin et al., 2024).

We performed an additional experiment studying the joint impact of LoRA and BDIA when fine-tuning GPT2 M for NLG. Since the validation performance is our primary concern, the LoRA+BDIA training procedure was implemented without online back-propagation and quantization. Differently from Subsection 5.1, $\{\gamma_k\}_{k=1}^{K-1}$ in the LoRA+BDIA procedure were randomly drawn from $\{0.5, 0, -0.5\}$ per training sample.

Next we briefly discuss the setup for LoRA's $\alpha$ parameter in the LoRA+BDIA procedure. In order for the BDIA approach to have a significant effect on the performance, it is recommended to use large values for the $\alpha$ parameter. This is because a large $\alpha$ value in LoRA resembles the full fine-tuning scenario to a certain extent. It is known from the main paper that BDIA is designed to average every two consecutive integration approximations in a random manner via $\{\gamma_k\}_{k=1}^{K-1}$. A large $\alpha$ parameter allows the averaging operations in BDIA to have a substantial effect. In the experiment, we tested a range of configurations with $\alpha \in \{512, 1024\}$ and LoRA rank $\in \{32, 128\}$.

Fig. 8 visualizes the training and validation performance for both LoRA and LoRA+BDIA. Similarly to the results in the main paper, the training loss of LoRA+BDIA is higher than that of LoRA alone due to the regularization effect of BDIA. On the other hand, the validation loss of LoRA+BDIA is lower than that of LoRA at the end of training, which is consistent with our expectation.

Table 6 summarizes the quality of the generated texts for four training setups evaluated with five different metrics. It is clear

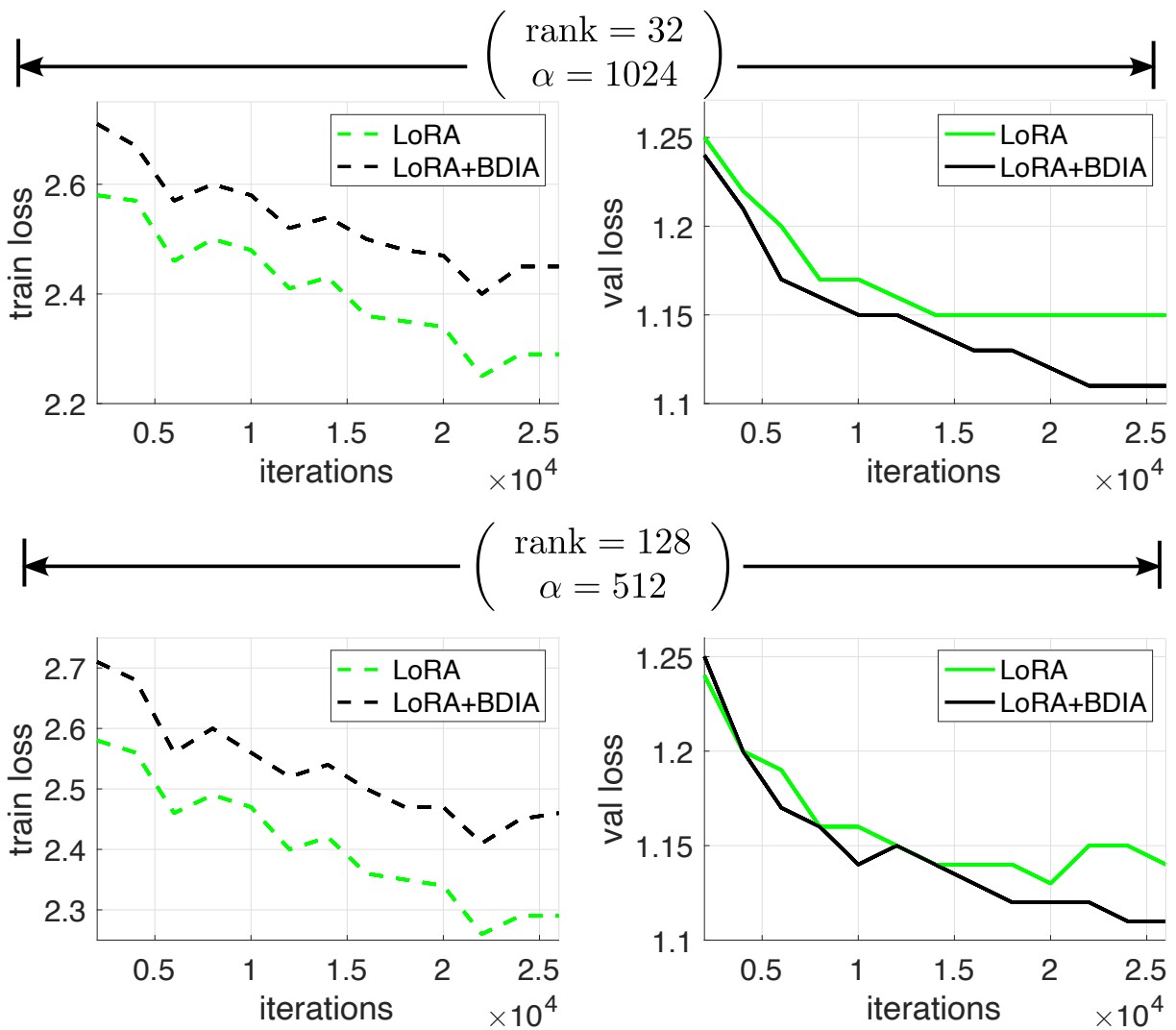

*Figure 8.* Performance comparison of LoRA nad LoRA+BDIA when fine-tuning GPT2 Medium for the E2E challenge. $\{\gamma_k\}_{k=1}^{K-1}$ in the BDIA training procedure were randomly drawn from $\{0.5, 0, -0.5\}$ per training sample.

*Table 6.* Performance of LoRA and LoRA+BDIA for fine-tuning GPT-2 medium (M) on the E2E NLG Challenge. For all metrics, higher is better. $\{\gamma_k\}_{k=1}^{K-1}$ in the BDIA training procedure were randomly drawn from $\{0.5, 0, -0.5\}$ per training sample.

| | Method | BLEU | NIST | MET | ROUGE-L | CIDEr |
|---|---|---|---|---|---|---|
| $\begin{pmatrix} \text{rank} = 32 \\ \alpha = 1024 \end{pmatrix}$ | LoRA | 68.8 | 8.66 | **46.8** | 71.6 | 2.46 |
| | LoRA+BDIA | **69.9** | **8.79** | 46.7 | **71.9** | **2.53** |
| $\begin{pmatrix} \text{rank} = 128 \\ \alpha = 512 \end{pmatrix}$ | LoRA | 68.0 | 8.58 | 46.1 | 70.9 | 2.49 |
| | LoRA+BDIA | **69.0** | **8.70** | **46.5** | **71.5** | **2.52** |

that for all tested scenarios, the LoRA+BDIA training procedure outperforms the LoRA procedure for almost all metrics. This suggests that the BDIA training approach can help LoRA in fine-tuning transformer-based LLMs.

## F. On Training BDIA-GPT2

In this experiment, we train BDIA-GPT2 on the openwebtext dataset utilizing the 4th github link of Table 7. Our primary objective for this task is to find out if BDIA can help to alleviate the over-fitting issue of GPT2 (we omit the phase "nano" for simplicity) for a very small training dataset. In doing so, we only took a small (i.e., 0.05%) subset from the entire dataset

when training the model. The performance of GPT2 was evaluated as a reference. Both models have 12 transformer blocks. We did not perform hyperparameter exploration and it is possible that the performance of BDIA-GPT2 would improve with such exploration.

Fig. 9 shows the training and validation curves for the two models. Similarly to Fig.4-5, BDIA-GPT2 exhibits a slower training speed than GPT2. Considering the validation performance, the two validation curves for the two models exhibit the over-fitting issue. GPT2 produces the lower validation loss in the middle of the training. However, at the end of training, the validation loss of BDIA-GPT2 is significantly lower than that of GPT2. This demonstrates that BDIA indeed alleviates the over-fitting issue of GPT2 for a very small training dataset.

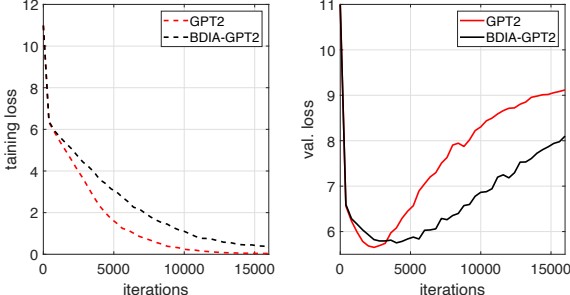

*Figure 9.* Performance comparison when training GPT2. $\{\gamma_k\}_{k=1}^{K-1}$ in the training procedure of BDIA-ViT were randomly drawn from $\{\pm 0.5\}$ per training sample.

## G. Repositories in the Experiments

*Table 7.* Repositories being used in the experiments

| | |
|---|---|
| image classification | https://github.com/kentaroy47/vision-transformers-cifar10 |
| natural language generation (NLG) | https://github.com/microsoft/LoRA/tree/main/examples/NLG |
| language translation | https://debuggercafe.com/language-translation-using-pytorch-transformer/ |
| nanoGPT | https://github.com/karpathy/nanoGPT |

