# OpenReview forum: "On Exact Bit-level Reversible Transformers Without Changing Architecture"
_ICML.cc/2025/Conference — ICML 2025 poster_

### Official Review · Reviewer_tmy4 · 2025-03-10

**Overall Recommendation:** 3

**Summary:**

The paper proposes BDIA-transformer, a novel approach combining the bidirectional integration approximation (BDIA) method and activation quantization to achieve exact bit-level reversibility in standard transformer architectures. This combination significantly reduces memory usage during training via online back-propagation, where activations are recomputed on-the-fly rather than stored. Additionally, random sampling of the hyperparameter $\gamma  \in ${-0.5, 0.5} introduces regularization that empirically improves generalization. Experiments on ViT-based image classification (CIFAR-10/100), GPT2-based language translation show BDIA-transformer achieving better validation performance and reduced training memory compared to baseline transformers.

**Claims And Evidence:**

Most claims concerning improved validation accuracy, reduced memory consumption, and generalization benefits are supported by experiments. While implicitly verified, it would be great to update Fig 2. to add the reconstruction errors when using a model following Equation 24 in order to confirm the validity of the exact byte inversion.

**Essential References Not Discussed:**

To my knowledge, the paper adequately cites and discusses relevant existing literature;

**Experimental Designs Or Analyses:**

The experimental designs appear valid, clearly illustrating the benefits of BDIA-transformer under limited data conditions. Nonetheless, an analysis of scenarios where model capacity, rather than data quantity, limits performance would help clarify the applicability and limitations of the proposed regularization method. While large-scale experiments are not expected, toy experiments could give a good picture of how the method trades model capacity to improve generalization.

**Methods And Evaluation Criteria:**

The methods and evaluation criteria generally make sense for the demonstrated applications. CIFAR10/100, GPT2 fine-tuning, and English-French translation tasks are adequate benchmarks to show memory efficiency and regularization effects. However, these memory considerations often arise in large-scale settings where the model's capacity is the limiting factor, but the experiments are done in a context where data is the limiting factor (overfitting setting).

**Other Comments Or Suggestions:**

- Clarify the theoretical underpinnings of the reversibility claim explicitly.
- Include sensitivity analysis regarding quantization precision.
- Consider additional experiments explicitly varying model size and data to highlight potential trade-offs.

**Other Strengths And Weaknesses:**

Strengths:
- Novel combination of reversible architectures with quantization for memory efficiency and regularization.
- Empirical validation across multiple popular tasks demonstrates general applicability.
- The fact that it minimally changes the architecture allows the use of this method for finetuning, which is a creative and powerful use case.

Weaknesses:
- Missing formal proofs regarding exact reversibility under quantization.
- No detailed exploration of how quantization precision (parameter l) impacts performance and memory. It would help to disentangle the benefits from reversibility and the benefits from quantization.
- Limited experimental settings that explicitly test the potential for underfitting or performance degradation when model capacity is constrained.

**Questions For Authors:**

- I am curious about how sensitive your method is to the choice of quantization precision (l). Especially how this precision affects both performance and reversibility?
- I don't see particular counter-indications to applying this strategy to blocks other than attention. Can similar observations be made for convolutional networks?

**Relation To Broader Scientific Literature:**

The paper positions itself appropriately within literature on reversible neural networks and activation quantization, clearly differentiating its unique combination of these two ideas from existing approaches like RevNet and RevViT.

**Theoretical Claims:**

No formal proofs were provided to substantiate theoretical claims, particularly regarding exact reversibility under quantization, leaving this as a significant gap.

---

> ### Author Rebuttal · Authors · 2025-03-31
>
> The authors thank all four reviewers for their appreciation of the novelty, simplicity, and effectiveness of our BDIA training technique for transformers. Notably, reviewer 7zwY states that __the novelty and effectiveness of the approach make this a strong paper__. Reviewer tmy4 states __minimally changes the architecture allows the use of this method for finetuning, which is a creative and powerful use case__. We would greatly appreciate the reviewers checking our detailed rebuttal responses and reconsidering their scores for our paper.
>
> __(1) Clarify the theoretical underpinnings of the reversibility claim explicitly.__
>
> In the revision, we will introduce a proposition and a proof in Subsection 4.3 to explicitly formulate the reversibility claim. Basically, in the proof, we will show that with quantization and the side information $s_{k-1}$, $x_{k-1}$ can be reconstructed exactly from $x_k$ and $x_{k+1}$ in online back-propagation,
> which only involves simple algebra. The side information $s_{k-1}$ essentially accounts for the binary quantization loss introduced by the configuration $\gamma_k\in \\{\pm 0.5\\}$.
>
> __(2) Include sensitivity analysis regarding quantization precision__
>
> As also suggested by reviewers 7ZwY and 8yrY,  we fully agree that conducting the ablation study on the impact of different quantization levels would be really helpful to understand the behavior of BDIA. A complete sensitivity analysis would require additional investigation, but we have performed additional experiments for testing the performance of both ViT and BDIA-ViT for the quantization levels of $\\{5, 0, -2\\}$ over CIFAR10.  In principle, the quantization level of $l=-2$ provides very coarse quantization effect and should lead to degraded performance.
>
> As shown in the table below, when the quantization level goes from 5 to -2, the performance of ViT and BDIA-ViT decreases as expected. One can also observe from the table that when $l\in \\{5, 0\\}$, BDIA-ViT performs consistently better than ViT. For the case of $l=-2$, ViT performs slightly better than BDIA-ViT. This likely is because the very coarse quantization dominates the performance and negatively affects the BDIA-training technique.
>
>  |quantization level |5|0|-2|
> | ------------------------ | ----- | --- | --- |
>  |ViT|87.90|87.76|__84.31__|
> |BDIA-ViT | __89.50__ |__88.47__ |84.16|
>
> __(3) Regarding the relationship between  quantization precision and reversibility.__
>
> In principle, the quantization precision is independent of the reversibility of BDIA-transformer. The reversibility of BDIA-transformer is guaranteed if the side information is stored for any particular quantization level. As we mentioned above, the quantization precision has a noticable impact on the validation performance. We will clarify the above point in the revision.
>
>
> __(4) Consider additional experiments explicitly varying model size and data to highlight potential trade-offs__
>
> Many thanks for the comments. We have conducted additional experiments by varying the number of transformer blocks in ViT and BDIA-ViT when trained over CIFAR10. As shown in the table below, BDIA-ViT performs consistently better than ViT for different numbers of transformer blocks.  We can also observe that increasing the number of transformer blocks does not lead to better validation performance for a fixed dataset. Neural ``scaling" law suggest that the model size should scale along with the increasing size of the dataset in a log-log relationship.
>
> We are not sure if we understand your suggestion on varying data for the additional experiments. It would be great if you could further elaborate on the above point so that we can perform appropriate additional experiments.
>
>
>
> | number of transformer blocks | 6|10|
> |-----------------------------------|----|----|
> |ViT|88.15 |88.02|
> |BDIA-ViT|__89.10__|__89.08__|
>
>
>
> __(5) Regarding the applicability of BDIA to other DNN blocks__
>
> Reviewer 8yrY suggested us to consider other transformer variants, such as linear transformer. Yes, we agree with your observation, although of course it would be necessary to empirically verify this.
> We have evaluated the BDIA technique for training vision-linformer over CIFAR10 by adopting the open source from https://github.com/pranavphoenix/VisionXformer/tree/main. Basically, we replaced the attention block in ViT by the linear attention block from the above open source in doing the experiment. The table below shows that BDIA again improves the validation accuracy considerably.
>
>
> Due to very limited time for rebuttal, we don't have time to test other DNN blocks. It would be of great interest to further investigate the effectiveness of BDIA for other DNN blocks such as longformer, Mamba blocks, and convolutional networks in the future. We hope our work provides a motivation for the above research directions.
>
> ||CIFAR10|
> |---------------------|------|
> |vision-linformer|84.16|
> |BDIA-vision-linformer|__86.6__|

---

> > ### Comment · Reviewer_tmy4 · 2025-04-03
> >
> > Points 1,3 and 5: I do not doubt that the statement is true, but having such proof would help the reader understand the roots of the method. Especially, a reader (like me) would have understood that the questions in 3) and 5) are a bit irrelevant by looking at such proof's assumptions.
> >
> > Point 2: I would like to apologize for the wrong phrasing of my question: by performance and memory, I meant performance in terms of speed (as quantization is often used to speed up training). Without any code nor any appendix to look at, I assumed that the quantization was done through a framework like [torch.amp](https://pytorch.org/docs/stable/amp.html), which made me think that quantization directly induces a speedup and a memory gain (by switching from float32 to bfloat16 or float8). However, after a more careful reading of eq.21 it seems like only the activations at the beginning/end of each transformer block are quantized (but the computation within each block is still full precision as $Q[(1-\gamma)x_k + (1+ \gamma)h_k(x_k)] \neq Q[(1-\gamma)x_k] + Q[(1+ \gamma)h_k(x_k)]$). Since those activations are not stored in online backprop, I did not understand that quantization has no impact on training speed/memory consumption.
> >
> > Point 4: My question was around the observation that experiments are done in the context where you want to mitigate overfitting; however, large-scale settings are more about mitigating underfitting. The second regime can be explored by reducing the model size.
> >
> > I will keep my rating to weak accept; I would not oppose this paper's acceptance as I do agree with the novelty and effectiveness of the approach. On the other side, I will not raise my grade because the *paper writing* falls short of the strong potential of *his idea*: with the total absence of appendices and code, the reader is forced to guess important aspects of the paper (discussion about point 2 is a good example of such guess). In short, I feel bad seeing this paper being weakly accepted, whereas this paper would be strongly accepted with sufficient appendices and available code.

---

> > > ### Author Response · Authors · 2025-04-05
> > >
> > > Many thanks for the valuable feedback for us to improve the paper.
> > >
> > > (1) For your information, we have applied BDIA in training resnet networks over CIFAR10 and obtained positive results again in terms of validation performance.
> > >
> > > || validation accuracy |
> > > |----------------------------------------------------|------|
> > > |ResNet with 15 residual blocks|92.93|
> > > |BDIA-ResNet with 15 residual blocks|__93.61__|
> > >
> > > (2) As suggested by reviewer BqRX, we have also studied the impact of dropout rates $\\{0.0, 0.1, 0.2\\}$ on the validation performance of BDIA-ViT over CIFAR10.   The results for dropout rate of 0.1 are taken from the paper. The table below demonstrates that the dropout and BDIA can work together to improve the validation performance.
> > >
> > > |dropout|0.0|0.0|
> > > |---------|----|---|
> > > | |ViT|BDIA-ViT|
> > > |CIFAR10|86.27|__89.20__|
> > > |CIFAR100|59.13|__64.45__|
> > > |dropout|0.1|0.1|
> > > ||ViT|BDIA-ViT|
> > > |CIFAR10|88.15|__89.10__|
> > > |CIFAR100|61.86|__66.09__|
> > > |dropout|0.2|0.2|
> > > ||ViT|BDIA-ViT|
> > > |CIFAR10|87.24|__88.22__|
> > > |CIFAR100|61.68|__64.24__|

---

### Official Review · Reviewer_8yrY · 2025-03-13

**Overall Recommendation:** 3

**Summary:**

The paper introduces the BDIA-transformer, a novel reversible transformer that maintains the standard transformer architecture during inference while leveraging a technique called bidirectional integration approximation (BDIA) for reversibility. The key idea is to treat each transformer block as an Euler integration approximation for solving an ordinary differential equation (ODE) and then apply BDIA to achieve reversibility. Additionally, activation quantization is incorporated to ensure exact bit-level reversibility. Experimental results on image classification, natural language generation, and language translation tasks show that BDIA-transformers outperform conventional transformers in validation accuracy while significantly reducing training memory.

**Claims And Evidence:**

The paper claims that BDIA-transformers achieve exact bit-level reversibility without altering the standard transformer architecture during inference. The proposed method reduces memory consumption in training while improving validation accuracy through model regularization. Experimental evidence supports these claims, showing improved validation performance across multiple tasks and datasets. The memory savings are attributed to online back-propagation enabled by reversibility, and the performance gain is linked to the ensemble effect of ODE solvers induced by the stochastic nature of the hyper-parameter. The claim that BDIA-transformers outperform RevViT is also supported by empirical results, though the performance gain varies across datasets.

**Essential References Not Discussed:**

A deeper discussion on recent advances in memory-efficient transformer training techniques would strengthen the paper quality. Notably, works on linear attention mechanisms and memory-efficient self-attention could provide alternative perspectives on reducing memory consumption.

**Experimental Designs Or Analyses:**

Experimental results compare the proposed model against baseline transformers and RevViT across various tasks. Multiple datasets and evaluation metrics are adopted. The training setup, involving multiple experimental repetitions, mitigates the effect of randomness. The inclusion of ablation studies exploring the impact of different hyper-parameters and quantization settings provides additional insights into the model's behavior.

**Methods And Evaluation Criteria:**

The evaluation is conducted on tasks including image classification, natural language generation, and language translation. The metrics used include validation accuracy, training loss, and peak memory consumption. The use of CIFAR-10, CIFAR-100, and the E2E dataset provides diverse benchmarks to assess the generalization of the proposed method. The evaluation criteria are appropriate for demonstrating the advantages of memory efficiency and performance improvements in BDIA-transformers compared to standard transformers and RevViT.

**Other Comments Or Suggestions:**

- Can you clarify the impact of different quantization levels on performance?

- A discussion on the trade-off between memory savings and computational overhead would be helpful.

**Other Strengths And Weaknesses:**

Strengths:

- Novel use of BDIA in transformers, achieving exact bit-level reversibility without modifying the inference architecture.

- Significant memory savings and improved validation performance.

- Thorough experimental evaluation across diverse tasks. Insightful ablation studies on the hyper-parameter and activation quantization.

Weaknesses:

- Increased memory consumption due to storing lightweight side information.

- Dependence on quantization introduces complexity in implementation.

**Questions For Authors:**

1. How does the choice of quantization level affect the model's performance and memory savings?

2. Can BDIA be applied to other transformer variants, such as linear transformers or long-form transformers?

3. What are the potential limitations of BDIA when scaling to large models with more than 100B parameters?

**Relation To Broader Scientific Literature:**

The paper builds on prior work in reversible neural networks and ODE-based modeling, extending these ideas to transformers. References to RevNet, neural ODEs, and RevViT contextualize the contributions within the broader literature. The connection to diffusion models and integration approximations further enriches the theoretical foundation.

**Theoretical Claims:**

The theoretical claim  is supported by the mathematical formulation of the update expressions. The derivation of BDIA update rules and the use of activation quantization ensure that each forward and backward step can be losslessly reconstructed. The use of lightweight side information per transformer block to counteract binary quantization loss further strengthens the theoretical soundness.

---

> ### Author Rebuttal · Authors · 2025-03-31
>
> The authors thank all four reviewers for their appreciation of the novelty, simplicity, and effectiveness of our BDIA training technique for transformers. Notably, reviewer 7zwY states that __the novelty and effectiveness of the approach make this a strong paper__. Reviewer tmy4 states __minimally changes the architecture allows the use of this method for finetuning, which is a creative and powerful use case__. We would greatly appreciate the reviewers checking our detailed rebuttal responses and reconsidering their scores for our paper.
>
> __(1) Regarding the impact of different quantization levels on performance.__
>
> We have performed additional experiments to test the performance of both ViT and BDIA-ViT for the quantization levels of $\{5, 0, -2\}$ over CIFAR10.  In principle, the quantization level $l=-2$ provides very coarse quantization effect and should lead to degraded performance.
>
> As shown in the table below, when the quantization level goes from 5 to -2, the performance of ViT and BDIA-ViT decreases as expected.  One can also observe from the table that when $l=5, 0$, BDIA-ViT performs consistently better than ViT. For the case of $l=-2$, ViT performs slightly better than BDIA-ViT. This could be because the very coarse quantization dominates the performance and negatively affects the BDIA-training technique.
>
>  |quantization level |5|0|-2|
> | ------------------------ | ----- | --- | --- |
>  |ViT|87.90|87.76|__84.31__|
> |BDIA-ViT | __89.50__ |__88.47__ |84.16|
>
> __(2) Regarding a discussion on the trade-off between memory savings and computational overhead__
>
> In the revision, we will add a table indicating the average running time per epoch for training ViT, BDIA-ViT with online backpropagation, and BDIA-ViT without online backpropagation over CIFAR10. We will then explain that BDIA-ViT with online backpropagation saves training memory and improves validation performance at the cost of slightly longer training time than ViT. We will also include additional figures in the revision showing the convergence curves in terms of wall-clock time.
>
> __(3) Regarding the impact of quantization level on memory savings__
>
> Firstly, we note that the reversibility of BDIA-transformer is guaranteed when the side information is stored for any quantization level in the forward pass of the model. The memory for the side information  is fixed for a particular transformer model and is independent of the quantization level.
>
> Secondly, in principle, when a coarse quantization is performed to the intermediate activation values across the transformer blocks, less memory is needed for the quantized activation values. Therefore, BDIA-transformer saves more memory when a coarser quantization is performed. However, the quantization cannot be too coarse as the performance of BDIA-transformer will be degraded, as shown in the above table.
>
> __(4) Can BDIA be applied to other transformer variants?__
>
> Many thanks for the suggestions. We have evaluated BDIA for training vision-linformer over CIFAR10 by adopting the open source from https://github.com/pranavphoenix/VisionXformer/tree/main. Basically, we replaced the attention block in ViT by the linear attention block from the above open source in doing the experiment. The table below shows that BDIA again improves the validation accuracy.
>
> We believe that BDIA can be applied to other transformer variants. Due to limited time for rebuttal, we did not have time to test other attention variants.  It would be of great interest to further investigate the effectiveness of BDIA for those variants in the future.
>
> ||CIFAR10|
> |---------------------|------|
> |vision-linformer|84.16|
> |BDIA-vision-linformer|__86.6__|
>
> __(5)  What are the potential limitations of BDIA when scaling to large models?__
>
> These large models are rumored to require thousands of GPUs to train, and we will not have access to such resources, so our response is speculative. Our hypothesis is that a large model usually has many transformer blocks. For example, Google states that GPT-4 has 120 transformer blocks. In principle, the regularization impact of the BDIA training technique increases along with the increasing number of transformer blocks since the number of ODE solvers parameterized by $\\{\gamma_{k}\\}_{k=1}^{K-1}$ in BDIA increases exponentially with $K$. We hypothesize that the validation performance of the large model would be improved noticeably by using the BDIA training technique.
>
> One thing that is  not yet clear to us is if BDIA is compatible directly from an engineering viewpoint with the distributed data parallel (DDP) framework, which is widely used for training large transformer models across many GPUs. Additional engineering work might be required for solving the above potential issue.
>
> __(6) Regarding literature review on linear attention mechanisms and memory-efficient self-attention__
>
> We will update the introduction accordingly to reflect the works you mentioned.

---

### Official Review · Reviewer_7ZwY · 2025-03-17

**Overall Recommendation:** 4

**Summary:**

This paper introduces BDIA-transformer, an exact bit-level reversible transformer that maintains the standard architecture for inference while reducing memory consumption during training. The approach adopts bidirectional integration approximation (BDIA), allowing the authors to consider each transformer block as an exactly reversible flow (once corrected by a 1-bit stored value at each activation layer) -- a framing that is theoretically and practically important in the study of ordinary differential equations. This reversibility provides a mechanism to recompute activations during the backwards pass when training the DNNs.

More specifically, the key contributions of the paper are: (1) introducing a random hyperparameter $\gamma \in [-0.5,0.5]$ per transformer block per training sample, which regularizes the model by effectively training an ensemble of ODE solvers; and (2) implementing activation quantization to enable exact bit-level reversibility. During inference, $\gamma$ is set to its expectation (zero), which reduces the architecture to a standard transformer with only activation quantization applied.

**Claims And Evidence:**

The authors claim their approach enables exact bit-level reversibility. This is supported by the mathematical formulation, though direct empirical verification of this property is somewhat limited in the experimental section.

The paper claims the core inference architecture is preserved by their technique. This is adequately supported by equations demonstrating how $E(\gamma)=0$ results in a standard transformer update with only quantization added.

The paper claims BDIA-transformer reduces memory consumption during training. This is supported by memory measurements shown in Table 1, which demonstrates reduced peak memory usage compared to standard transformers.

The authors claim BDIA-transformer outperforms conventional transformers due to regularization effects. The evidence in Figures 3-6 and Tables 1-3 supports this, showing improved validation performance across different tasks.

**Essential References Not Discussed:**

None.

**Experimental Designs Or Analyses:**

It would be helpful to see comparisons against other memory-saving techniques beyond RevViT.

**Methods And Evaluation Criteria:**

The methods and evaluation criteria (application domains, baseline architectures, and $\gamma$ ablation study) are generally sufficient.  Given the computational claims of the paper, metrics demonstrating such overheads would make for a more comprehensive analysis.

**Other Comments Or Suggestions:**

The paper would benefit from a more thorough analysis of the computational overhead introduced by BDIA-transformer. While memory savings are quantified, training time comparisons would provide a more complete picture.

The paper mentions that RevViT has better memory efficiency than BDIA-transformer. A more nuanced discussion of this trade-off would strengthen the paper.

The ablation studies could be expanded to include analysis of the impact of different quantization levels (beyond the 9-bit quantization considered).

**Other Strengths And Weaknesses:**

The novelty and demonstrated effectiveness of the approach make this a strong paper.

One weakness is the lack of detailed computational overhead analysis.

**Questions For Authors:**

Could you elaborate on how BDIA-transformers might be integrated with other memory-saving techniques like gradient checkpointing, selective activation recomputation, and activation offloading? Are there specific combinations you've explored or would recommend?

How sensitive is the method to different quantization levels, and what are the performance trade-offs?

What specific challenges did you encounter when implementing BDIA for different transformer architectures, and how might these insights guide others implementing your approach?

**Relation To Broader Scientific Literature:**

The authors provide a good overview of reversible neural networks, and ODE-based neural networks.  The Bidirectional Integration Approximation is reviewed.  Well known quantization strategies are mentioned.

**Theoretical Claims:**

Yes.  The claims are derivational in nature and are reasonable.

---

> ### Author Rebuttal · Authors · 2025-03-31
>
> The authors thank all four reviewers for their appreciation of the novelty, simplicity, and effectiveness of our BDIA training technique for transformers. Notably, reviewer 7zwY states that __the novelty and effectiveness of the approach make this a strong paper__. Reviewer tmy4 states __minimally changes the architecture allows the use of this method for finetuning, which is a creative and powerful use case__. We would greatly appreciate the reviewers checking our detailed rebuttal responses and reconsidering their scores for our paper.
>
> __(1) Regarding the computational overhead introduced by BDIA-transformer.__
>
> In the revision we will add a table indicating the average running time per epoch for training ViT, BDIA-ViT with online backpropagation, and BDIA-ViT without online backpropagation over CIFAR10.  We will then explain that BDIA-ViT with online backpropagation saves training memory and improves validation performance at the cost of slightly longer training time than ViT. We will also include additional figures in the revision showing the convergence curves in terms of wall-clock time.
>
> __(2) Regarding a discussion of RevViT having better memory efficiency than BDIA-transformer__
>
> We will elaborate in the paragraph at lines 369-376 in the experimental section regarding the benefit of RevViT with respect to memory efficiency.
>
> __(3) Regarding the impact of different quantization levels__
>
> As also suggested by reviewers 8yrY and tmy4, we fully agree that conducting the ablation study on the impact of different quantization levels would be really helpful for understanding the behaviors of the BDIA training technique. We have performed additional experiments to test the performance of both ViT and BDIA-ViT for the quantization levels of $\{5, 0, -2\}$ over CIFAR10.  In principle, a quantization level of $l=-2$ provides a very coarse quantization effect and should lead to degraded performance.
>
> As shown in the table below, when the quantization level goes from 5 until -2, the performance of ViT and BDIA-ViT decreases as expected. For $l=-2$, there is a large performance drop due to the very coarse quantization effect. One can also observe from the table that when $l\in \\{5, 0\\}$, BDIA-ViT performs consistently and significantly better than ViT. For the extreme case of $l=-2$, ViT performs slightly better than BDIA-ViT. This likely is because the very coarse quantization dominates the performance and negatively affects the BDIA-training technique.
>
>  |quantization level |5|0|-2|
> | ------------------------ | ----- | --- | --- |
>  |ViT|87.90|87.76|__84.31__|
> |BDIA-ViT | __89.50__ |__88.47__ |84.16|
>
>
> __(4) Regarding integrating BDIA-transformers with other memory-saving techniques__
>
> There is one scenario where the BDIA training technique can be nicely combined with gradient checkpointing. We note that some transformer architectures may have special bottleneck blocks working as dimensionality transition (similar to the bottleneck blocks in ResNet). In the above case, a certain number of the transformer blocks of the same dimensionality are present between every two bottlenecks for efficient representation learning.
>
> In the above scenario, one cannot directly perform online backpropagation from the very top transformer block to the bottom one due to dimensionality inconsistency. In this case, one can apply gradient checkpointing to handle the bottleneck blocks, and apply BDIA to do online backpropogation for those  transformer blocks of the same dimensionality between every two bottlenecks.
>
>
> __(5)  What specific challenges did you encounter when implementing BDIA for different transformer architectures?__
>
> Many thanks for the comment. We will release our source code once the paper is accepted so that other researchers can easily implement the BDIA training technique based on our source code.
>
> From our experience,  BDIA without online backpropagation can be easily implemented for different transformer architectures by following the update expressions in the paper. The challenging part is to realize online back-propagation for BDIA, which we implemented by modifying the open source code for RevViT. Researchers who only want to improve the validation performance of a transformer can simply ignore the online-backpropagation part of BDIA.
>
>
> __(6) Regarding comparisons against other memory-saving techniques beyond RevViT.__
>
> Many thanks for the comment. We plan to investigate the performance of the middle-point reversible transformer in the revision, of which the update expression is given by
>
> $x_{k+1}= x_{k-1}+ 2 \\left[\\textrm{FFN}_k( {x}_k+\\textrm{Atten}_k({x}_k)) + \\textrm{Atten}_k({x}_k) \\right],  \\quad k=1,\ldots, K-1.$
>
> It is expected that BDIA-transformer will still perform better than the middle-point reversible transformer because the latter one does not introduce any regularization into the neural network.

---

### Official Review · Reviewer_BqRX · 2025-03-17

**Overall Recommendation:** 3

**Summary:**

The paper proposes a novel type of reversible transformers with the aim to reduce the memory during training. To this end, this work treats each transformer block as the Euler integration approximation in a manner similar to Neural ODEs. There are two main contributions. Firstly, the authors borrow a technique from recent works on diffusion inversion for round-trip image editing, which involves bidirectional integration approximation. This approximation introduces a hyperparameter  $\gamma$. The authors propose selecting $\gamma$ randomly either -0.5 or 0.5 for each training sample and training block. Consequently, the training can be viewed as an ensemble of ODE solvers. This regularization led to observed improvements on validation data. Secondly, to ensure reversibility, the authors propose performing activation quantization while storing side information. This approach is validated on small datasets involving image classification, machine translation, and language modeling.

----
Update after the rebuttal:
----

I have carefully read the authors' rebuttal as well as the comments from the other reviewers. After consideration, I have decided to increase the score. I hope the authors integrate the suggestions from the reviewers into the next version of the paper and also release the code.

**Claims And Evidence:**

There is a problem. Although the paper provides detailed mathematical derivations, they appear to be more closely aligned with concepts from residual networks (ResNets) rather than being specifically tailored to transformers.

**Essential References Not Discussed:**

No

**Experimental Designs Or Analyses:**

Yes. I checked all the experiments.

**Methods And Evaluation Criteria:**

Yes

**Other Comments Or Suggestions:**

No

**Other Strengths And Weaknesses:**

Strengths:
- The paper addresses an important and timely problem: reducing the memory consumption during the training of transformers, which is particularly relevant given the current widespread use of transformer models.
- The proposed idea is compelling as it retains the original architecture of transformers. This stands in contrast to existing approaches that typically involve modifications to the transformer architecture.

Weaknesses:
- The paper is difficult to follow. For example, the abstract is too long.
- The reproducibility is low as there is no source codes or pseudo codes or detailed algorithms.
- Although the paper includes thorough mathematical derivations, these seem to be more aligned with concepts from residual networks (ResNets) rather than focusing specifically on transformers. Notably, in equation (4), the authors treat the combined attention and feed-forward network modules as a residual term, resulting in derivations similar to those found in NeuralODEs with ResNets. However, these modules are key differentiators in transformer architectures compared to other models.
- The experiments mainly consider small datasets or relies on toy examples for transformers.

**Questions For Authors:**

- In figure 1, and in line 287, how did the authors integrate into standard transformers?
- In figure 2, the authors should show the reconstruction errors w.r.t. the proposed method using quantization and side information. Otherwise, it is not clear the effectiveness of these tricks.
- The authors should compare experimentally the proposed methods against vanilla transformers applied with dropout.
- Although the authors show the memory gains, they should show the convergences in terms of wall-clock time to see better the computational complexity introduced by the proposed method.

**Relation To Broader Scientific Literature:**

It is a relevant contribution to the literature as the investigated problem is timely: reducing the memory consumption during the training of transformers.

**Theoretical Claims:**

Yes

---

> ### Author Rebuttal · Authors · 2025-03-31
>
> The authors thank all four reviewers for their appreciation of the novelty, simplicity, and effectiveness of our BDIA training technique for transformers. Notably, reviewer 7zwY states that __the novelty and effectiveness of the approach make this a strong paper__. Reviewer tmy4 states __minimally changes the architecture allows the use of this method for finetuning, which is a creative and powerful use case__. We would greatly appreciate the reviewers checking our detailed rebuttal responses and reconsidering their scores for our paper.
>
> __(1)  Regarding the paper readability__
>
> Based on  comments from all four reviewers, we will make the following changes in the revision to improve readability:
>
> (a)  We will shorten the abstract to make it more concise.
>
> (b) As suggested by reviewer tmy4,  we will modify Subsection 4.3 by including a proposition as well as a proof for that proposition. The proposition will relate to the reversibility of BDIA with quantization and side information.
>
> (c) The pseudocode for the BDIA training method will be included in the paper to enable the readers to easily understand the method.
>
> (d) A few additional ablation studies will be included in the revision. In particular, we will study the  impact of dropout rates, number of transformer blocks, and quantization levels on the validation performance of BDIA-ViT. We note that all the above ablation studies demonstrate that BDIA-ViT performs consistently better than ViT in terms of validation accuracy. We will also present the computational overhead of BDIA-ViT with online back-propagation in the revision.
>
> __(2)  Regarding the reproducibility and pseudocodes of BDIA__
>
> In the revision, we will include the pseudocode for the BDIA training method. We will also release the source code after the paper is accepted so that other researchers will be able to adapt our source code to different transformer-based tasks.
>
> __(3) Although the paper includes thorough mathematical derivations, these seem to be more aligned with concepts from residual networks (ResNets) rather than focusing specifically on transformers.__
>
>
> It is true that the derivations take a unified viewpoint, i.e., that architectures with skip connections can be regarded from a numerical integration point of view. Components of the particular schemes (ResNet, NeuralODE, Transformer, etc.) are simply regarded as __black-box__ functions in this view. This unified viewpoint makes BDIA applicable to not only transformers but also other schemes.  The reason we focus on transformers in the paper is that transformers are commonly employed in models that (i) are very large and (ii) are the state-of-the-art in terms of performance. More-over, related methods have not been studied in the context of transformers. To reflect the well-motivated comment by the reviewer we will further strengthen the interpretation of our approach for transformers specifically in the final version of the paper.
>
> __(4) Regarding the small datasets in the experiments__
>
> Many thanks for the comment. Firstly, as commented by reviewer tmy4, the BDIA training technique could be a good candidate for fine-tuning transformer-based models because BDIA retains the standard architecture in the inference procedure. Fine-tuning a pre-trained model often works with a dataset of small or intermediate size. Our experiment for fine-tuning GPT2 medium using a small dataset in the paper shows that BDIA produces promising results.
>
> Secondly, even though the considered datasets in our experiments are not very large, all three different training tasks (i.e., image classification, NLG, language translation) in our paper show that the BDIA training technique improves the validation performance of transformer consistently and considerably, demonstrating the potential applicability of the BDIA training technique in different applications.
>
> __(5) Regarding impact of dropout__
>
> Firstly, to clarify, the experimental results of Table 1 and 2 in the paper are obtained by setting the dropout rate to be 0.1. We will revise the paper accordingly to clarify the ambiguity.
>
> Secondly, we conducted additional experiments by setting the dropout rate to be 0.0 and 0.2.  The table below demonstrates that the dropout and BDIA can work together to improve the validation performance.
>
> |dropout | 0.0 |0.0 |
> |--------------|---------------|----------|
> | |ViT|BDIA-ViT|
> | CIFAR10| 86.27|__89.20__|
> |CIFAR100| 59.13|__64.45__|
> |dropout |0.2 | 0.2 |
> | |ViT|BDIA-ViT|
> | CIFAR10| 87.24|__88.22__|
> |CIFAR100|61.68|__64.24__|
>
> __(6) Regarding figure 2__
>
> We will update figure 2 to demonstrate that the reconstruction error for BDIA with  quantization and side information is zero. That is, __lossless__ online back-propagation is guaranteed with BDIA thanks to the quantization and side information.
>
> __(7) Regarding figure 1__
>
> We will revise the caption of figure 1 to better explain how it is generated.

---

### Decision · Program_Chairs · 2025-05-01

**Decision:**

Accept (poster)

**Comment:**

This manuscript targets the problem of reducing memory use during training of (large scale) transformer models. This is accomplished through the use of a bidirectional integration approximation (BDIA). This is complemented by quantization.

The claims of the manuscript are well supported by the experiments (though the scale is, somewhat necessarily, limited). The reviewers like the underlying ideas behind the work and there does seem to be real potential for impact. Nevertheless, reviewers did identify that the presentation could be improved and more concrete theoretical claims would strengthen the manuscript (which are supposedly forthcoming as minor revisions). The interesting and potentially impactful core ideas in the manuscript paired with its good empirical evaluation justify my recommendation.